# PKR activation-induced mitochondrial dysfunction in HIV-transgenic mice with nephropathy

Teruhiko Yoshida[1]*[†], Khun Zaw Latt[1][†], Avi Z Rosenberg[2], Briana A Santo[3], Komuraiah Myakala[4], Yu Ishimoto[5], Yongmei Zhao[6], Shashi Shrivastav[1], Bryce A Jones[4], Xiaoping Yang[2], Xiaoxin X Wang[4], Vincent M Tutino[3], Pinaki Sarder[3,7], Moshe Levi[4], Koji Okamoto[1,8], Cheryl A Winkler[6], Jeffrey B Kopp[1]

[1]Kidney Disease Section, Kidney Diseases Branch, NIDDK, NIH, Bethesda, United States; [2]Department of Pathology, Johns Hopkins Medical Institutions, Baltimore, United States; [3]Department of Pathology and Anatomical Sciences, Jacobs School of Medicine & Biomedical Sciences, University at Buffalo, Buffalo, United States; [4]Department of Biochemistry and Molecular & Cellular Biology, Georgetown University, Washington, United States; [5]Polycystic Kidney Disease Section, Kidney Diseases Branch, NIDDK, NIH, Bethesda, United States; [6]Frederick National Laboratory for Cancer Research, NCI, NIH, Frederick, United States; [7]College of Medicine, University of Florida, Gainesville, United States; [8]Nephrology Endocrinology and Vascular Medicine, Tohoku University Hospital, Sendai, Japan

*For correspondence: yoshidateruhiko@g.ecc.u-tokyo.ac.jp

[†]These authors contributed equally to this work

Competing interest: The authors declare that no competing interests exist.

**Abstract** HIV disease remains prevalent in the USA and chronic kidney disease remains a major cause of morbidity in HIV-1-positive patients. Host double-stranded RNA (dsRNA)-activated protein kinase (PKR) is a sensor for viral dsRNA, including HIV-1. We show that PKR inhibition by compound C16 ameliorates the HIV-associated nephropathy (HIVAN) kidney phenotype in the Tg26 transgenic mouse model, with reversal of mitochondrial dysfunction. Combined analysis of single-nucleus RNA-seq and bulk RNA-seq data revealed that oxidative phosphorylation was one of the most downregulated pathways and identified signal transducer and activator of transcription (STAT3) as a potential mediating factor. We identified in Tg26 mice a novel proximal tubular cell cluster enriched in mitochondrial transcripts. Podocytes showed high levels of HIV-1 gene expression and dysregulation of cytoskeleton-related genes, and these cells dedifferentiated. In injured proximal tubules, cell-cell interaction analysis indicated activation of the pro-fibrogenic PKR-STAT3-platelet-derived growth factor (PDGF)-D pathway. These findings suggest that PKR inhibition and mitochondrial rescue are potential novel therapeutic approaches for HIVAN.

## eLife assessment

This study presents **valuable** new insights into a HIV-associated nephropathy (HIVAN) kidney phenotype in the Tg26 transgenic mouse model, and delineates the kidney cell types that express HIV genes and are injured in these HIV-transgenic mice. A series of **compelling** experiments demonstrated that PKR inhibition can ameliorate HIVAN with reversal of mitochondrial dysfunction (mainly confined to endothelial cells), a prominent feature shared in other kidney diseases. The data support that inhibition of PKR and mitochondrial dysfunction has potential clinical significance for HIVAN.

## Introduction

The World Health Organization (WHO) reported that in 2020, ~1.5 million people became newly HIV positive, ~37 million people were living with HIV, and ~680,000 people globally died from HIV-related causes (*Fauci, 2022*). HIV remains prevalent in the USA (*Prevention CfDCa, 2019*) and even more highly prevalent in sub-Saharan Africa, especially in East and Southern Africa (*Dwyer-Lindgren et al., 2019*). Chronic kidney disease (CKD) remains a major co-morbidity in HIV-positive individuals, even with availability of combined anti-retroviral therapy (*Palau et al., 2018*). The most severe form of CKD in persons with untreated or undertreated HIV infection is HIV-associated nephropathy (HIVAN), collapsing glomerulopathy. Therapy for HIVAN includes combined anti-retroviral therapy, coupled with renin-angiotensin-aldosterone system blockade, to prevent CKD progression (*Menez et al., 2018*; *Yahaya et al., 2013*).

The Tg26 transgenic is a widely used HIVAN model. The HIV-1 transgene expresses in kidney cells and the model manifests histological features similar to those of human HIVAN, including collapsing glomerulopathy, microtubular dilatation, and interstitial fibrosis (*Bruggeman et al., 1997*). The mice develop progressive renal dysfunction and progress to terminal uremia (*Dickie et al., 1991*; *Kopp et al., 1992*).

Diverse therapeutic approaches are effective in Tg26 kidney disease, including inhibitors of mammalian target of rapamycin (*Kumar et al., 2010*; *Yadav et al., 2012*; *Rai et al., 2013*), Notch inhibition (*Sharma et al., 2013*; *Puri et al., 2019*), renin-angiotensin system inhibition (*Zhong et al., 2013*; *Rai et al., 2014*), cyclin-dependent kinase inhibition (*Gherardi et al., 2004*), sirtuin1 agonist or overexpression (*Wang et al., 2020*), STAT3 activation reduction (*Feng et al., 2009*; *Gu et al., 2013*), retinoic acid receptor agonist (*Ratnam et al., 2011*), and NF-kB inhibition (*Zhang et al., 2012*). Recent reports have identified several injury pathways, including NLRP3 (*Haque et al., 2016*) and mitochondrial dysfunction (*Bryant et al., 2018*; *Katuri et al., 2019*). Bulk multi-omics approaches using mRNA microarrays and protein-DNA arrays (*Jin et al., 2012*) identified homeo-domain interacting protein kinase-2 (HIPK2) as a regulator of Tg26 renal pathology; this was confirmed in recent reports (*Liu et al., 2017*; *Xiao et al., 2020*). An mRNA microarray approach characterized bulk transcriptional profiles during progressive renal disease (*Fan et al., 2014*). While these reports have provided novel insights into tissue transcriptional dynamics, studies have not been performed at single-cell or single-nucleus resolution.

Double-stranded RNA (dsRNA)-activated protein kinase (PKR) is a sensor for dsRNA and is activated in response to viral infections, including HIV-1. In the USA and globally, HIV remains an important problem that disproportionately affects marginalized groups, including the Black/African-American community (*Fauci, 2022*). *APOL1* risk variants, exclusively present in individuals with recent sub-Saharan ancestry, damage podocytes through various mechanisms including PKR activation by *APOL1* mRNA (*Okamoto et al., 2018*). The PKR-inhibiting oxoindole/imidazole compound C16 is beneficial in neuroinflammatory disease models (*Tronel et al., 2014*; *Ingrand et al., 2007*).

We hypothesized that PKR activation is a mechanistic pathway shared by HIV- and APOL1-mediated nephropathies, considering the high odds ratio for HIVAN among African Americans (OR 29) and South Africans (OR 89) carrying two *APOL1* risk alleles (*Kasembeli et al., 2015*). We investigated the effects of PKR inhibition in the Tg26 HIVAN mouse model, which expresses HIV regulatory and accessory genes. We used single-nucleus and bulk RNA-seq methods to identify transcriptional changes between study groups and to uncover associated molecular mechanisms in an unbiased fashion.

## Results

### PKR inhibition ameliorates kidney injury in Tg26 mice

The trans-activating regions RNA in the HIV-1 long terminal repeats at the 5' and 3' ends of the HIV genome form dsRNA structures and induce PKR autophosphorylation, thereby activating PKR (*Carpick et al., 1997*). As PKR is a potent driver of many stress response pathways, including translational shutdown, apoptosis, inflammation, and metabolism, we hypothesized that PKR inhibition by the PKR-specific oxoindole/imidazole inhibitor C16 might rescue kidney injury in Tg26 mice. We administered C16 to Tg26 and wild-type (WT) mice from 6 to 12 weeks of age, and we evaluated the kidney phenotype. C16-treated Tg26 mice had lower serum creatinine (*Figure 1A*), reduced albuminuria (*Figure 1B and C*), and reduced pPKR abundance (*Figure 1D–G*) compared to control Tg26 mice.

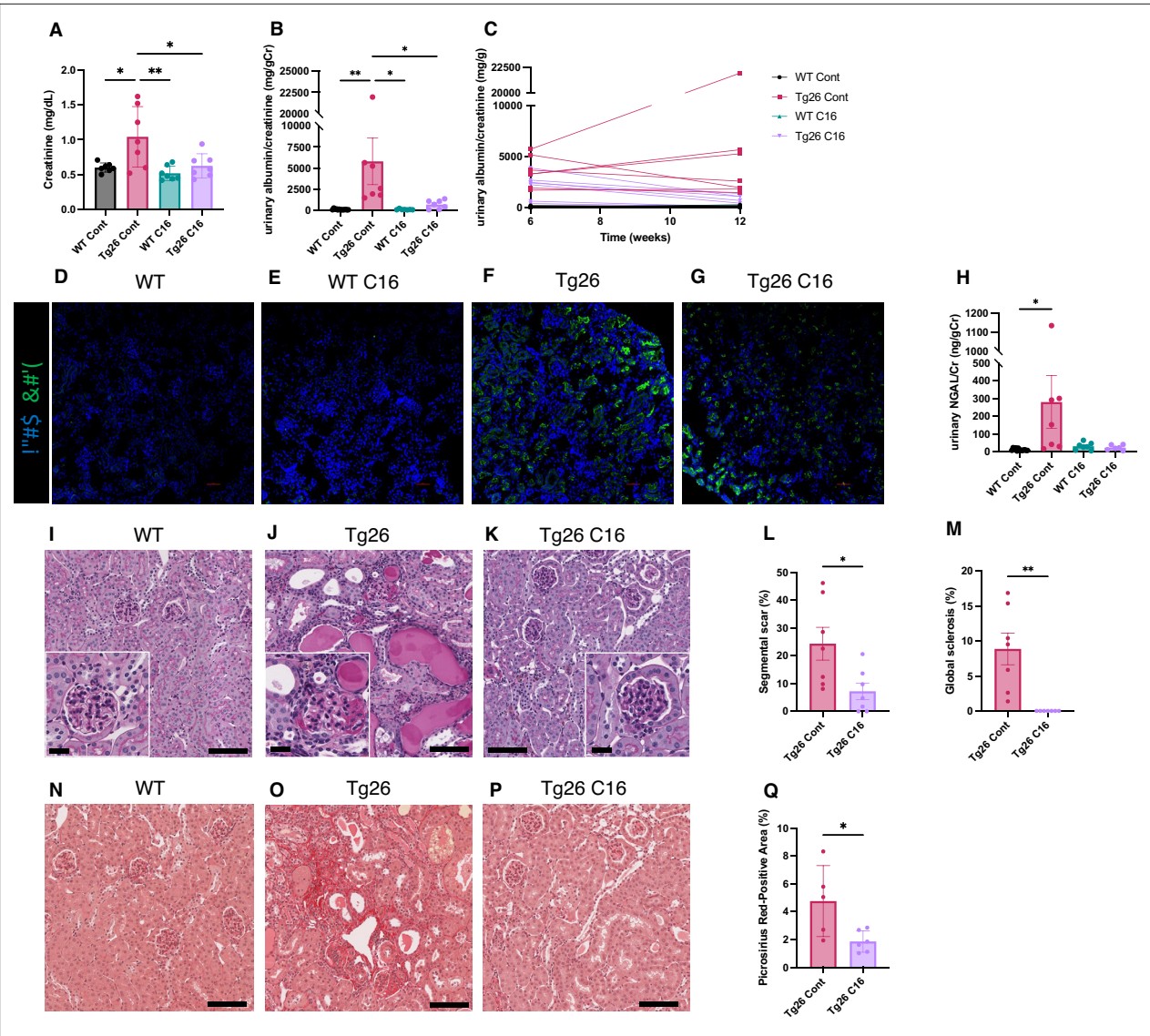

**Figure 1.** PKR inhibition by C16 ameliorates Tg26 mice kidney phenotype. (**A–C**) Shown are the following: plasma creatinine (mg/dl, mean with SD), urinary albumin-to-creatinine ratio (mg/g creatinine, mean with SEM), urinary albumin-to-creatinine ratio (mg/g creatinine) of 6 and 12 weeks of age. (**D–G**) Immunofluorescent images showed PKR activation by detecting pPKR in Tg26 mouse kidney. pPKR was inhibited by C16 treatments. (**H**) Shown is urinary neutrophil gelatinase-associated lipocalin (NGAL)-to-creatinine ratio (ng/g creatinine, mean with SEM). (**I–K**) Representative PAS staining images of wild-type (WT), Tg26, and C16-treated Tg26 kidney. (**L, M**) Quantitative analysis of glomeruli for segmental scarring and global sclerosis (mean with SEM). (**N–P**) Representative Picrosirius Red staining images of WT, Tg26, and C16-treated Tg26 kidney. (**Q**) Quantitative analysis of Picrosirius Red staining area (mean with SD). (one-way ANOVA (**A, B, H**), t-test (**L, M, Q**); *, p<0.05; **, p<0.01; scale bars are 50 μm).

Further, C16 treatment reduced urinary excretion of the kidney injury marker neutrophil gelatinase-associated lipocalin (NGAL) (*Figure 1H*).

At 12 weeks of age, there was less glomerulosclerosis and of microtubular dilatation in C16-treated Tg26 mice (*Figure 1I–K*). Histomorphological quantification confirmed that C16 treatment reduced glomerular injury, assessed as segmental glomerulosclerosis and global glomerulosclerosis (*Figure 1L and M*), and as fibrosis extent, quantified by Picrosirius Red staining (*Figure 1N–Q*).

## Combination of single-nucleus RNA-seq and bulk RNA-seq to profile transcriptomic changes in Tg26 mice

To investigate molecular mechanisms in Tg26 kidney and the effect of PKR inhibition, we conducted bulk RNA-seq of kidney cortex from the four groups (WT, WT treated with C16, Tg26, and Tg26

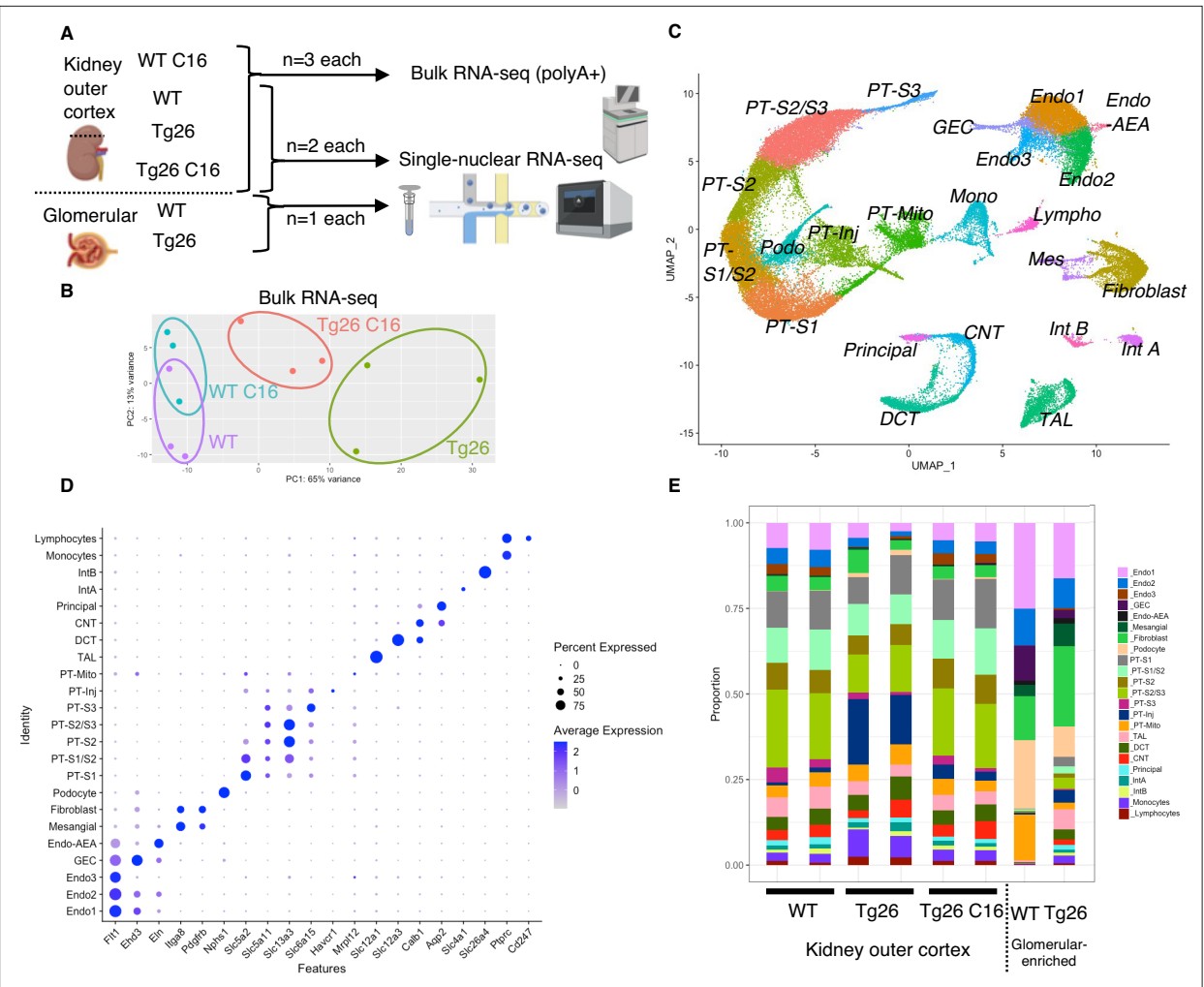

**Figure 2.** Overview of bulk RNA-seq and single-nucleus RNA-seq experiments. (**A**) Shown is the workflow of the bulk RNA-seq and single-nucleus RNA-seq experiments. (**B**) Principal component analysis plot of bulk RNA-seq results. (**C**) Uniform manifold approximation and projection (UMAP) plot of single-nuclear RNA-seq data from 8 samples, 56,976 nuclei, showing 23 clusters. (**D**) Shown is a dot plot of 23 marker genes, each characteristic for the cluster. (**E**) Shown is the ratio of nuclei grouped to each cluster in each sample.

The online version of this article includes the following figure supplement(s) for figure 2:

**Figure supplement 1.** Pre-processing of single-nuclear RNA-seq data.

**Figure supplement 2.** Additional plots of single-nuclear RNA-seq data.

treated with C16; n=3 each) and single-nucleus RNA-seq of kidney cortex from six samples (WT, Tg26, and Tg26 treated with C16; n=2 each) and of two samples of glomeruli (WT and Tg26; n=1 each) to enrich for glomerular cells (*Figure 2A*). Bulk RNA-seq data clustered well by treatment groups in a principal component analysis plot (*Figure 2B*). Single-nucleus RNA-seq profiled 56,976 nuclei. We identified 23-cell clusters, including a novel cell type (PT-Mito, proximal tubule cell cluster with a higher expression level of mitochondrial genes compared to adjacent cells), as shown as a uniform manifold approximation and projection (UMAP) plot (*Figure 2C*, *Figure 2—figure supplement 1*). Marker genes for each cluster used for annotation are shown in *Figure 2D*. Taking advantage of unbiased clustering in single-nucleus RNA-seq, we tabulated cell numbers from each mouse kidney, using kidney cortex and glomeruli (*Figure 2E*, *Figure 2—figure supplement 2A*). The gene encoding PKR, *Eif2ak2*, was expressed in glomeruli and all tubular segments (*Figure 2—figure supplement 2B*).

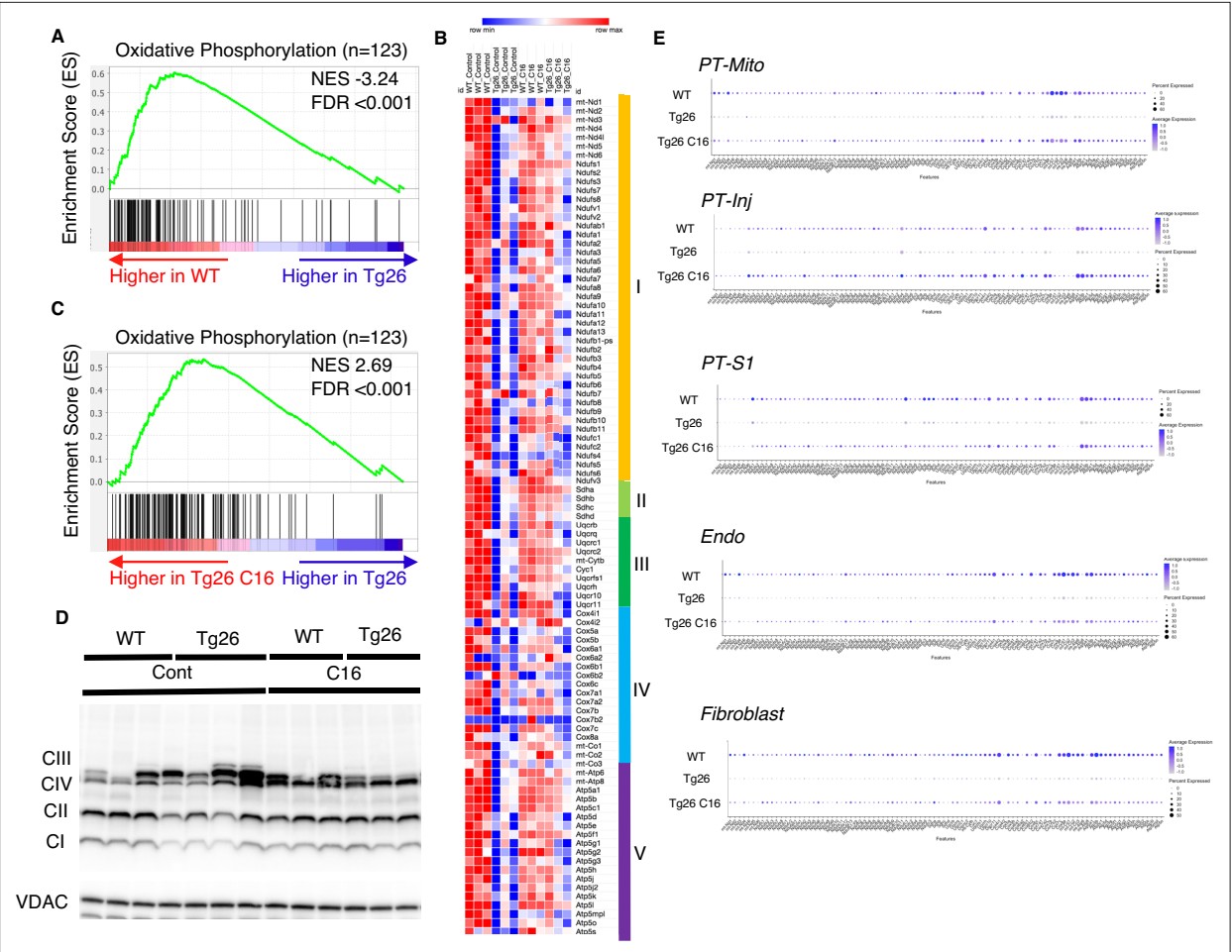

**Figure 3.** Oxidative phosphorylation genes are downregulated in Tg26 mice and downregulation is reversed by PKR inhibition using C16. (**A**) Shown is the enrichment plot of oxidative phosphorylation pathway based on bulk mRNA-seq comparing Tg26 and wild-type (WT). (**B**) Shown is the enrichment plot of oxidative phosphorylation pathway based on bulk mRNA-seq comparing C16-treated Tg26 and Tg26. (**C**) Heatmap of expressed genes in oxidative phosphorylation pathway (n=123), based on data from bulk mRNA-seq. (**D**) Western blot to identify mitochondrial subunits CI through CIV and VDAC. (**E**) Dot plot showing expression of oxidative phosphorylation pathway genes in PT-Mito, PT-S1, PT-Inj, Endo, and Fibroblast cluster by snRNA-seq.

The online version of this article includes the following source data and figure supplement(s) for figure 3:

**Source data 1.** Uncropped and labeled gels for *Figure 3D*.

**Source data 2.** Raw unedited gels for *Figure 3D*.

**Figure supplement 1.** Additional data about mitochondria and S phase score of podocytes.

## Downregulation of mitochondrial genes in Tg26 kidneys

Gene set enrichment analysis (GSEA) results of bulk RNA-seq showed that mitochondrial-related pathways were the most downregulated pathway in Tg26 mice when compared with WT mice, suggesting that mitochondrial gene transcription (for both nuclear-genome and mitochondrial genome-encoded genes) was significantly downregulated (*Figure 3A*). Expression levels of specific mitochondrial genes were assessed (*Figure 3B*) and showed that C16 treatment reversed the downregulation of these mitochondrial genes (*Figure 3B and C*). Western blot analyses were consistent with bulk RNA-seq results showing lower abundance of mitochondrial complex I and II in Tg26 kidney and this was restored to normal by C16 treatment (*Figure 3D*, *Figure 3—figure supplement 1A–D*). Mitochondrial DNA copy numbers were decreased in Tg26 mice, but C16 treatment did not reverse these changes (*Figure 3—figure supplement 1E and F*).

These findings suggest that PKR inhibition by C16 rescued transcriptional downregulation of both nuclear-encoded and mitochondrial-encoded mitochondrial genes, but the rescue was not through

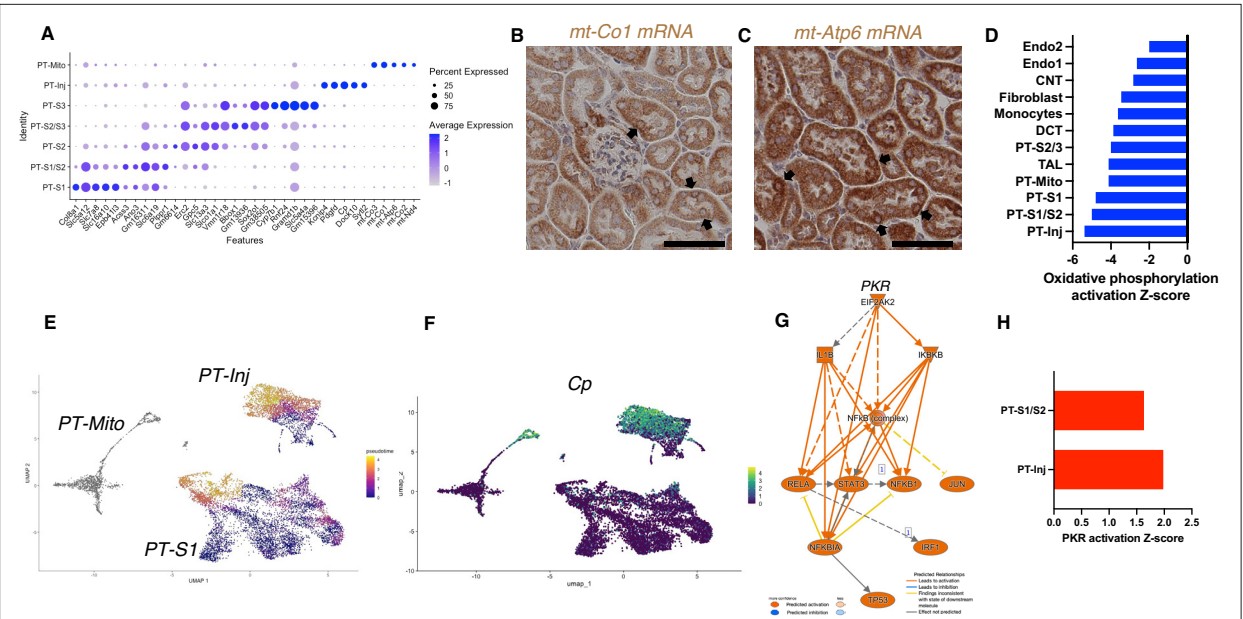

**Figure 4.** PT-Mito and PT-Inj cluster characterization. (**A**) Shown are dot plots showing the top five marker genes in each of the PT clusters (PT-S1, PT-S1/S2, PT-S2, PT-S2/S3, PT-S3, PT-Inj, PT-Mito). (**B, C**) In situ hybridization of *mt-Co1* and *mt-Atp6* genes showed signals inside nuclei of wild-type (WT) mice (scale bars are 50 μm). (**D**) Shown is the activation Z-score of the oxidative phosphorylation pathway by pathway analysis of each cluster comparing Tg26 vs WT mice. (**E**) Trajectory analysis results including PT-S1, PT-Mito, PT-Inj from WT and Tg26 mice. (**F**) Ceruloplasmin (*Cp*) expression in the trajectory analysis plot. (**G**) PKR downstream pathway mapping by IPA comparing Tg26 vs WT by bulk mRNA-seq data. (**H**) Activation Z-score of PKR pathway by IPA in each cluster comparing Tg26 vs WT.

The online version of this article includes the following figure supplement(s) for figure 4:

**Figure supplement 1.** Additional in situ hybridization images.

**Figure supplement 2.** PT-Mito cluster detection of publicly available human kidney single-nuclear RNA-seq data (GSE131882).

**Figure supplement 3.** Single-nuclear RNA-seq data comparison with ischemic reperfusion injury model.

increased mitochondrial DNA copy number. Single-nucleus RNA-seq and subsequent pathway analysis showed that the majority of cell types, including mitochondrial proximal tubules (PT-Mito), PT-S1, injured PT (PT-Inj), and endothelial cells, manifested global downregulation of mitochondrial-expressed genes in Tg26 kidneys and that this decline was rescued by C16 treatment (*Figure 3E*).

## Novel proximal tubular cell cluster with high mitochondrial gene expression

Based on unbiased clustering, we identified a distinct proximal tubule cell cluster characterized by higher expression level of mitochondrial genes (PT-Mito) and consisting of 3.1–5.8% of kidney cortex cells (*Figure 2E*). We analyzed the seven proximal tubular cell clusters and found distinctive markers that distinguish each cluster (*Figure 4A*). Based on UMAP, this PT-Mito cluster was in proximity to proximal tubule-segment1 (PT-S1) and proximal tubule-segment2 (PT-S2) (*Figure 2C*). To confirm the presence of cells giving rise to this cluster, in situ hybridization (ISH) of mt-*Co1* and mt-*Atp6* were performed. We observed transcripts inside some nuclei in many putative PT-Mito segments with high expression of these genes (*Figure 4B and C*, *Figure 4—figure supplement 1A and B*). We also confirmed the existence of similar PT-Mito cluster in published human kidney single-nuclear RNA-seq data (*Wilson et al., 2019*) by the re-analysis of the original data (*Figure 4—figure supplement 2A–C*).

As the PT-Mito cluster was observed in both WT and Tg26 kidney in similar ratios, this new cluster could be a conserved tubular cell cluster, likely not previously reported due to mitochondrial gene filtering criteria implemented in single-nucleus RNA-seq analytic pipelines. Pathway analysis of snRNA-seq data showed that oxidative phosphorylation genes were downregulated in this PT-Mito cluster in Tg26 mice, compared to WT mice, with a Z-score of –4.123. These genes were also downregulated in other many cell types (*Figure 4D*). These findings suggested that mitochondrial dysfunction

represents a prominent mechanistic pathway that was dysregulated in majority of cell clusters in Tg26 mice, and that mitochondrial dysfunction was pronounced in the PT-Mito cluster.

## Proximal tubular cells with injury were more frequent in Tg26 mice

We identified a proximal tubular cell cluster, enriched for injury markers (PT-Inj), which was increased in cell number and percentage in Tg26 mice when compared with WT mice and C16-treated Tg26 mice. The gene expression profile of this cluster was comparable to profiles previously reported mouse models of acute kidney injury (*Kirita et al., 2020*) and kidney fibrosis (*Lu et al., 2021*). We compared gene expression in Tg26 mice with previously reported expression of marker genes in injured proximal tubules from snRNA-seq data obtained from a mouse ischemia-reperfusion injury model (*Figure 4—figure supplement 3A*). We found some overlap in these two models (Tg26, ischemia-reperfusion), and the higher expression of these genes in Tg26 mice was ameliorated with C16 treatment (*Figure 4—figure supplement 3B*). This finding suggested some overlap of pathological molecular pathways between HIV-associated tubular injury and ischemic injury, especially at later stages when fibrosis appears.

Pseudotime analysis suggested that injured proximal tubular epithelial cells likely originated from PT-S1 segments, suggesting proximal tubular damage in Tg26 mouse kidneys (*Figure 4E and F*). Differential gene expression analysis and IPA upstream analysis of bulk RNA-seq data from Tg26 and WT mouse kidneys showed activation of the PKR pathway (*Figure 4G*). The per cluster upstream analysis of snRNA-seq data showed the highest PKR pathway activation in PT-Inj (*Figure 4H*).

## PKR inhibition rescued Stat3 activation in Tg26 mice

PKR inhibition, acting to reduce inflammatory pathway activation, may influence other important mediators of HIVAN pathophysiology. To identify possible mediators, we used upstream analysis in IPA. We compared C16-treated WT vs WT, Tg26 vs WT, and C16-treated Tg26 vs Tg26, using differential gene expression analysis. All genes with multiple-testing adjusted p-values<0.05 were included in this analysis. An activated Z-score was calculated for each candidate upstream regulator. Candidate transcription factors were sorted in the order of triple Z-score, which was calculated by multiplying the three Z-scores (*Table 1*).

Stat3 and representative examples of Stat3-regulated gene expressions profiles are shown for each tissue comparison (*Figure 5A and B*). Interestingly, Stat3 was the most activated upstream regulator and is a well-characterized transcriptional regulator in kidney disease, including in the Tg26 mouse model (*Feng et al., 2009*). We confirmed Stat3 activation through phosphorylation by western blot showing increased phosphorylation (*Figure 5C and D*) and immunohistochemistry (*Figure 5E–G*). These data suggest that PKR inhibition may be therapeutic for Tg26 kidney disease, promoting deactivation of Stat3 and downstream inflammatory pathways. Based on single-nucleus RNA-seq and IPA upstream analysis, we confirmed that Stat3 was activated in the majority of cell types in Tg26 and

**Table 1.** Activation Z-scores from bulk RNA-seq data.

Activation Z-scores of transcription factors by upstream regulator analysis of Ingenuity Pathway Analysis. Triple Z-score was calculated by multiplying three Z-scores in each comparison.

| Upstream regulator | Activation Z-score WT C16 vs WT | Activation Z-score Tg26 vs WT | Activation Z-score Tg26 C16 vs Tg26 | Triple Z-score |
|---|---|---|---|---|
| STAT3 | –2.183 | 6.286 | –3.182 | 43.66 |
| FOXO1 | –3.618 | 3.991 | –2.697 | 38.94 |
| GLI1 | –2.335 | 5.517 | –2.936 | 37.82 |
| EPAS1 | –2.665 | 3.554 | –3.242 | 30.71 |
| SREBF1 | –3.013 | 3.672 | –2.63 | 29.10 |
| CREB1 | –2.325 | 4.181 | –2.916 | 28.35 |
| ATF4 | –2.606 | 4.445 | –2.028 | 23.49 |
| SMAD2 | –2.207 | 2.791 | –2.335 | 14.38 |

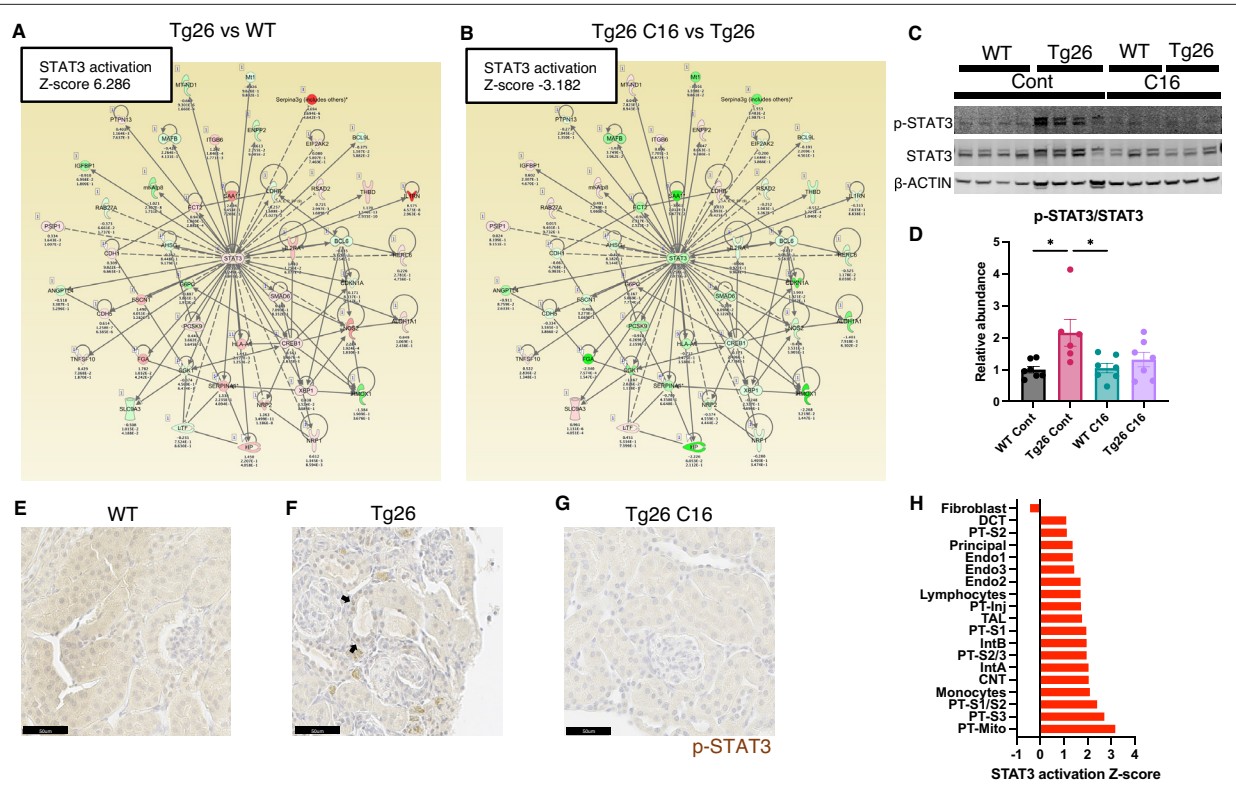

**Figure 5.** STAT3 activation downstream of PKR. (**A, B**) Shown is mapping of STAT3 regulating genes comparing Tg26 vs WT (Z-score 6.286), Tg26 C16 vs Tg26 (Z-score –3.182) by bulk mRNA-seq. Red color indicates upregulation and green color indicates downregulation, quantified as log2-fold change, p-value; and adjusted p-value by false discovery rate are shown. Solid line indicates known positive regulation, dotted line indicates known negative regulation. (**C**) Representative immunoblotting of phospho-STAT3, STAT3, and β-ACTIN. (**D**) Quantitative results of phospho-STAT3/STAT3 by immunoblotting (mean with SEM, one-way ANOVA; *, p<0.05). (**E–G**) Phospho-STAT3 immunostaining of mouse kidneys is shown (scale bars are 50 μm). Arrows indicate p-Stat3 detection in injured tubular cells. (**H**) STAT3 activation Z-score by upstream regulator analysis comparing Tg26 vs WT in each cluster by snRNA-seq.

especially in PT-Inj (*Figure 5H*). As Stat3 translocate to mitochondria and alter cell metabolism (*Xu et al., 2016*), Stat3 may also be an important mitochondrial regulator in the pathogenesis of HIVAN.

## PKR inhibition restored reduced mitochondrial respiration capacity in Tg26

These data indicated that mitochondrial-expressed gene transcription was inhibited in the kidney cortex of Tg26 mice and was rescued by PKR inhibition. To investigate mitochondrial functions in kidney, we prepared enriched glomerular and proximal tubular tissue extracts. These were tested for mitochondrial respiratory capacity using the Seahorse extracellular flux analyzer. The results showed reduced maximum respiration and spare respiratory capacity in Tg26 compared to WT mice (*Figure 6A–C*). Reduced respiratory capacity in Tg26 mice was rescued by C16 treatment, suggesting that PKR inhibition restored impaired mitochondrial respiration in Tg26 mice. Similarly, proximal tubular tissue ex vivo also showed lower maximum respiration and less spare respiratory capacity in Tg26 mice compared to WT mice (*Figure 6D–F*); both parameters were normalized by C16 treatment. Thus, PKR inhibition rescues mitochondrial dysfunction in both glomerular and proximal tubular cells.

## Podocytes in Tg26 mice showed high HIV-1 gene expression with podocyte loss

Although the Tg26 mouse is a well-characterized model of HIVAN, expression levels of each of the HIV-1 genes in single kidney cells has not been previously reported. Using single-nucleus RNA-seq, we annotated expression of each HIV-1 gene, with the aim to quantify HIV-1 gene expression at

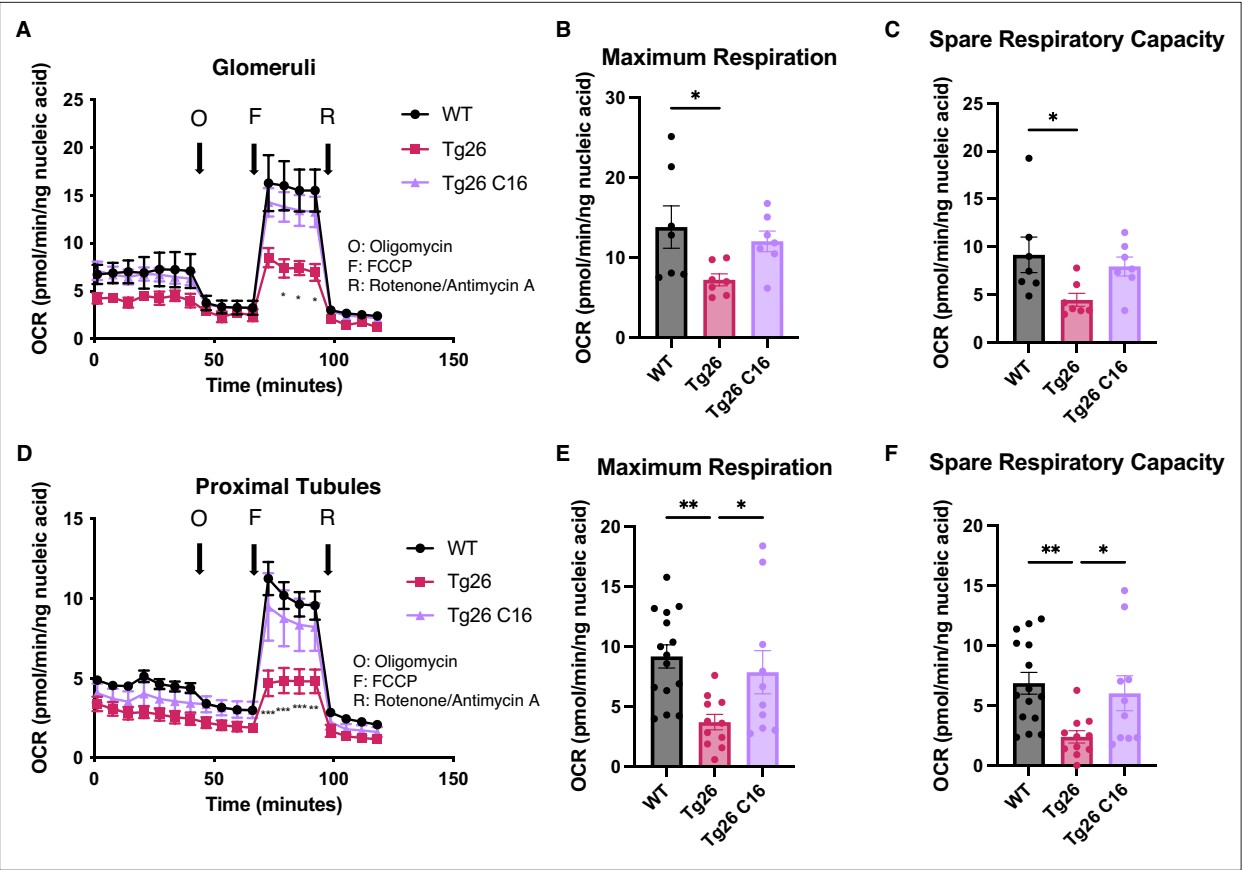

**Figure 6.** PKR inhibition with C16 reverses mitochondrial dysfunction in Tg26 glomeruli and proximal tubules. (**A**) Shown are oxygen consumption rate (OCR) measurements during cell mitochondrial stress testing, using extracted glomeruli from wild-type (WT), Tg26, and C16 Tg26 kidney tissue. (**B**) Maximum respiration rate was calculated by OCR measurements of glomeruli (mean with SEM). (**C**) Spare respiratory capacity was calculated by OCR measurements of glomeruli (mean with SEM). (**D**) Shown are OCR measurements during cell mitochondrial stress testing, using extracted proximal tubules from WT, Tg26, and C16 Tg26 kidney tissue. (**E**) Maximum respiration rate was calculated by OCR measurements of isolated proximal tubules (mean with SEM). (**F**) Spare respiratory capacity was calculated by OCR measurements of isolated proximal tubules (mean with SEM). (one-way ANOVA; *, $p < 0.05$; **, $p < 0.01$).

The online version of this article includes the following source data for figure 6:

**Source data 1.** Uncropped and labeled gels for *Figure 6C*.

**Source data 2.** Raw unedited gels for *Figure 6C*.

single-cell resolution (*Figure 7A*). Among all kidney cell types, HIV-1 genes were expressed at high levels in podocytes, consistent with the notion that HIV-1 gene expression is particularly injurious to podocytes. HIV-1 gene expression data from this study are consistent with the current understanding that *vpr* and *nef* are main contributors to the pathogenesis of HIVAN. Absolute podocyte loss in Tg26 mice, and podocyte recovery with C16 treatment, were also confirmed by p57 staining and podocyte estimation using the podometric analysis implemented in PodoCount (*Figure 7B–E*).

## Podocytes in Tg26 mice showed dedifferentiation and dysregulation of cytoskeleton-related pathways

Since podocytes in Tg26 mice expressed the HIV-1 genes *vpr* and *nef*, we investigated related pathways that were dysregulated in podocytes. The yield of podocytes from kidney cortex samples was relatively low and, therefore, we used snRNA-seq data from isolated glomeruli from WT and Tg26 mice, which enriched podocytes. Pseudotime analysis of these data showed progression in transcripts from WT podocytes to the majority of Tg26 podocytes, showing *nef* expression in most injured Tg26 podocytes (*Figure 7F and G*). Differential expression analysis and pathway analysis suggested that in Tg26 mice, Rho protein dissociation inhibitors (RHOGDI) signaling and corona virus pathogenesis

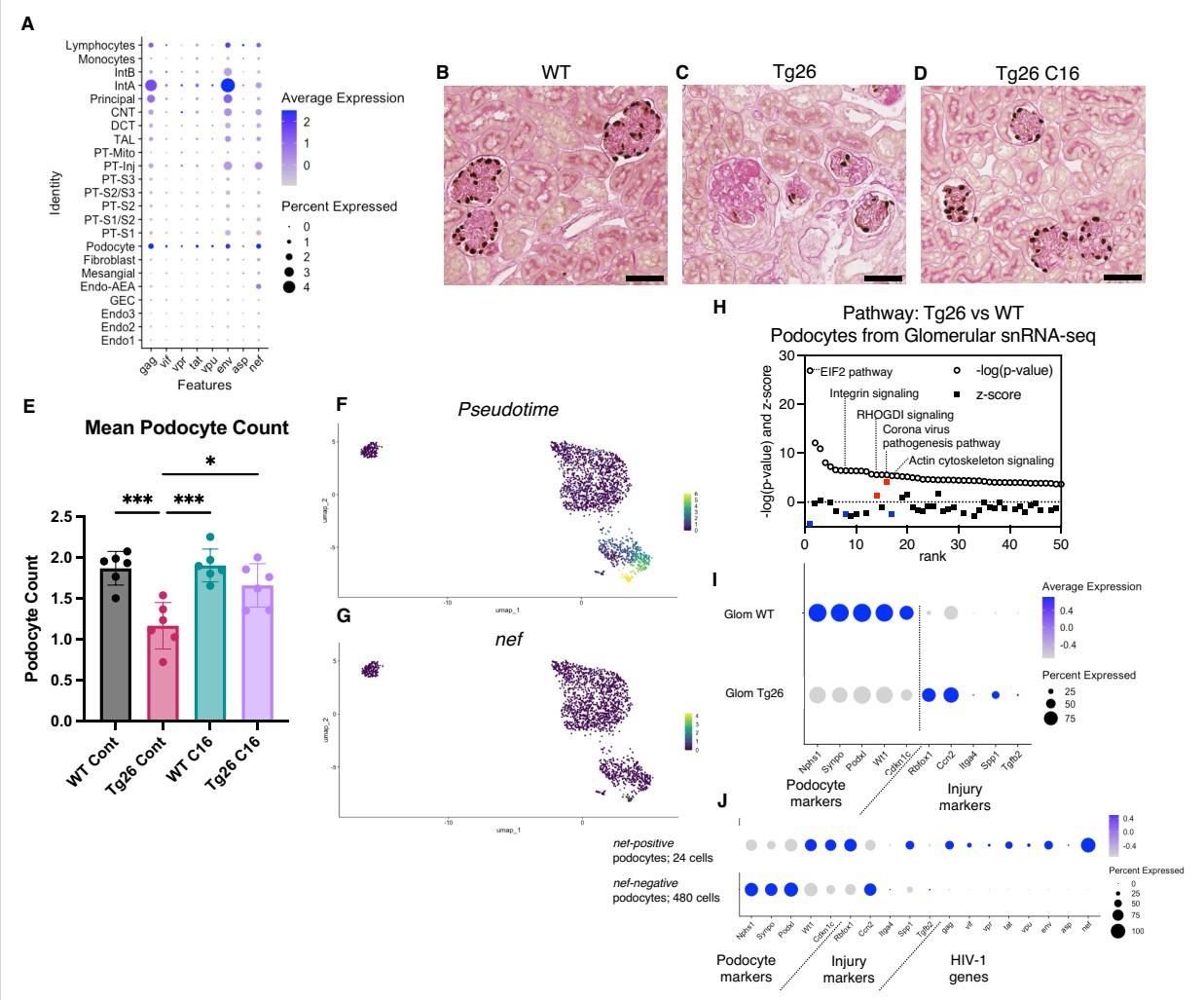

**Figure 7.** HIV-1 gene expression causes Tg26 podocytes dedifferentiation. (**A**) Shown is dot plot demonstrating HIV-1 gene expression levels in each cluster detected by snRNA-seq. (**B–D**) p57 staining of kidney showing podocyte loss and dedifferentiation (scale bars are 50 μm). (**E**) PodoCount analysis showed podocyte loss in Tg26 and was rescued by C16 (one-way ANOVA; *, p<0.05; ***, p<0.001). (**F**) Trajectory analysis of podocytes by snRNA-seq data from wild-type (WT), Tg26 glom samples. (**G**) Trajectory map showing *nef* expression. (**H**) Pathway analysis results by IPA comparing Tg26 vs WT using glomerular snRNA-seq data from podocyte cluster. (**I**) Dot plot showing podocyte marker genes and representative differentially expressed genes in podocytes by glomerular snRNA-seq. (**J**) Dot plot comparing expression of representative genes in glomerular Tg26 podocytes between *nef*-positive and *nef*-negative podocytes.

pathways were activated, while integrin and actin cytoskeleton signaling pathways were deactivated (***Figure 7H***).

These findings are consistent with the previous reports and with a common conceptual model in which podocyte dedifferentiation starts with cytoskeletal changes and progresses to cell detachment from the glomerular tuft (***Lu et al., 2008***; ***Husain et al., 2005***; ***Sunamoto et al., 2003***). We confirmed dedifferentiation of podocytes in Tg26 mice by showing downregulation of podocyte markers and reversal or prevention of dedifferentiation in Tg26 mice treated with C16 (***Figure 7I***). This was also demonstrated histologically by p57 staining, which was lost in Tg26 and then regained following C16 treatment (***Figure 7B–D***). We also found increased expression of *Ccn2* (encoding cellular communication network factor 2), an epithelial-mesenchymal transition marker, in Tg26 podocytes. This is consistent with the signatures that we previously reported in urine podocytes in a single-cell clinical study of FSGS patients (***Latt et al., 2022***). We also observed upregulation of *Rbfox1*, encoding RNA-binding fox-1 homolog-1, as candidate disease marker in Tg26 podocytes (***Figure 7I***). Rbfox1 also regulates key neuronal functions (***Carreira-Rosario et al., 2016***; ***Lee et al., 2016***).

Expression of integrin subunits *Itga4* and of *Spp1* (encoding osteopontin) and *Tgfb2* (encoding transforming growth factor β2) was also upregulated in Tg26 podocytes, indicating activation of cell adhesion and pro-fibrotic processes (*Figure 7I*). We compared gene expression in *nef*-positive podocytes (24 cells) and *nef-negative* podocytes (480 cells) from Tg26 glomerular data, to confirm the association between HIV-1 gene expression and transcriptomic changes. We found lower expression of canonical podocyte marker genes, including *Nphs1*, *Synpo*, together with higher expression of *Rbfox1* and *Spp1* described above (*Figure 7J*).

## Tg26 kidney cells showed active cell-cell interaction

To investigate pathological cell-cell interactions in Tg26 kidneys and the ameliorative effect of PKR inhibition, we performed cell-cell interaction analysis using the snRNA-seq data. The expression

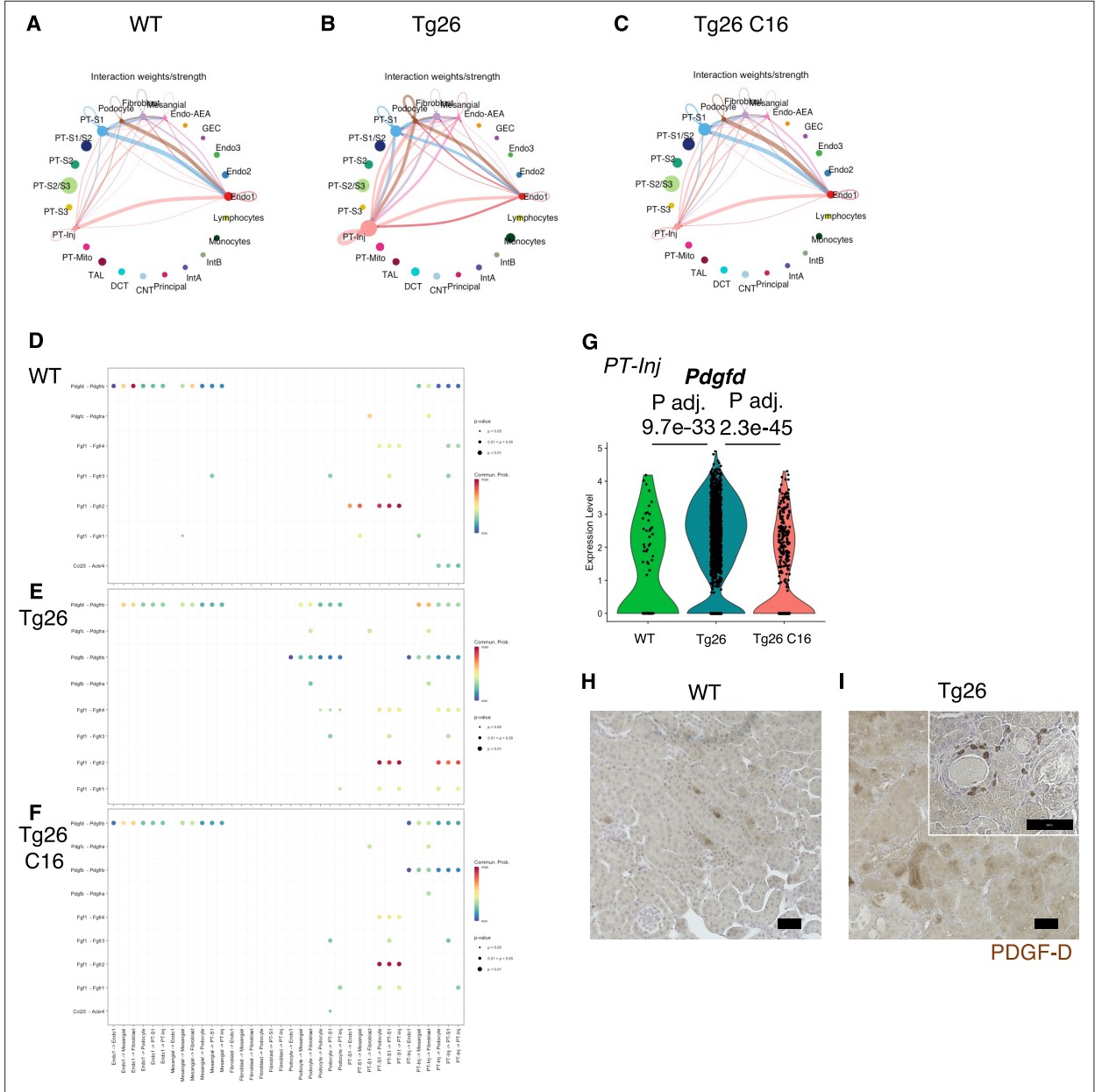

**Figure 8.** Cell-cell interaction analysis shows activated ligand-receptor interaction: platelet-derived growth factor (PDGF)-D-PDGFR-B pathway in Tg26. (**A–C**) Circles plot showing cell-cell interaction weight in each sample at the resolution of each identified cell cluster. (**D–F**) Shown are dot plots depicting results from cell-cell interaction analysis of wild-type (WT), Tg26, C16 Tg26 snRNA-seq data. (**G**) Violin plots showing Pdgfd expression levels in the PT-Inj cluster. (**H, I**) PDGF-D immunostaining of mouse kidneys is shown (scale bars are 50 μm).

heatmap suggested changes in cell-cell interaction in each sample at the resolution of each identified cell cluster (*Figure 8A–C*). We sought candidate cell types that might contribute to Tg26 pathology and found candidate ligand-receptor pairs (*Figure 8D–F*). For example, platelet-derived growth factor D (PDGF-D) was upregulated in PT-Inj in Tg26 mice and was downregulated by C16 treatment (*Figure 8G*). Further, PDGF-D may interact with PDGFR-B in fibroblasts. Immunohistochemistry demonstrated the presence of PDGF-D in the vicinity of dilated tubules (*Figure 8H and I*). As PDGF-D is known to be induced by STAT3, these findings identified a potential fibrogenic pathway triggered by PKR activation in PT-Inj.

## Discussion

This work, applying combined analysis of single-nucleus RNA-seq and bulk RNA-seq data, highlight the role of PKR activation in the Tg26 HIVAN mouse model and show mitochondrial dysfunction to be one of the most dysregulated pathways. Mitochondrial dysfunction in Tg26 mouse glomeruli and myocytes has been previously reported (*Bryant et al., 2018*; *Cheung et al., 2019*). The present study demonstrated by single-nucleus RNA-seq and Seahorse bioenergetics assay that mitochondrial dysfunction is induced in various cell types in Tg26 kidney, especially proximal tubular cells and endothelial cells. It is likely that HIVAN tubular pathology is mediated, at least in part, by mitochondrial dysfunction in proximal tubular cells, and that glomerular mitochondrial dysfunction is mainly confined to endothelial cells. Further, PKR activation may be one of several processes that contributes to mitochondrial dysfunction in HIVAN. A recent report using tubular cell-specific Vpr overexpression mouse model has shown that mitochondrial dysfunction is implicated in tubular injury. Our findings are congruent with this report (*Chen et al., 2023*).

Further, single-nucleus RNA-sequencing of Tg26 kidney cortex identified a novel mitochondrial gene-enriched proximal tubular cell population (PT-Mito). This cell population had not been previously recognized, likely due to filtering criteria routinely employed to reduced mitochondrial genome-encoded transcript levels in single-cell/single-nucleus datasets. It has been well established that a high percentage of mitochondrial transcripts captured in single-cell RNA-seq studies suggests stressed cells, and these cells are typically excluded from analysis. However, in single-nucleus RNA-seq datasets, mitochondrial gene percentage is generally low following the nuclear purification. Most nuclei captured have mitochondrial transcripts levels of <1% of the total transcripts captured. Therefore, we did not filter out mitochondrial transcripts entirely from the analysis but excluded nuclei with >20% mitochondrial transcripts (*Slyper et al., 2020*). By not filtering out mitochondrial genes in the analysis, we found dysregulation of these transcripts, providing clues to pathogenesis.

We further demonstrated by ISH the presence of mitochondrial transcripts in nuclei whose cells were identified in single-nuclei analysis of the same tissue samples. Although we cannot completely exclude the possibility of having captured some mitochondria during nuclear preparation, we confirmed expression of nuclear genes, such as *Gpx1* and *Gpx3*, which were also highly expressed in this PT-Mito cluster. A relatively high abundance of mitochondrial transcripts in the PT-Mito cluster may indicate the existence of mitochondrial transcripts that were transported into nuclei. This PT-Mito cluster might have remained previously undetected because of the similarity of its transcripts with that of those other proximal tubules and the lack of cell-type-specific markers. Considering the high mitochondrial gene expression levels and corresponding mitochondrial pathway dysregulation in Tg26 mice, this PT-Mito cluster may represent the most metabolically active cells, which are also the most highly vulnerable cell type in proximal tubules. Nevertheless, considering the downregulation of both nuclear and mitochondrial encoded genes involved in oxidative phosphorylation, mitochondrial dysfunction likely plays a major role in the pathology of HIVAN.

Another novel finding arose from investigating HIV-1 gene expression in the Tg26 mouse. Transcripts for all transgene-encoded HIV-1 genes were detected in all cell types examined, albeit some transcripts were present at low expression levels. This included transcripts for *nef* and *vpr* that are, according to current understanding, the main contributors to HIVAN (*Zuo et al., 2006*; *Rednor and Ross, 2018*). Podocytes showed the highest levels of transgene expression, which is consistent with the prominent pathology observed in podocytes. This finding suggests a shared but unknown mechanism in virus-related nephropathies with podocyte damage. One possible explanation for podocytes not showing overt mitochondrial gene dysregulation despite high HIV-1 gene expression (compared to proximal tubular cells and endothelial cells) is a possible tighter regulation of mitochondrial gene

expression in podocytes, as suggested by *Li et al., 2017*. With regard to S phase-specific genes (*Tirosh et al., 2016*), we did not find a consistent expression change in cells including podocytes from the glomerular sample (*Figure 3—figure supplement 1G*).

Further, we identified a putative activated pathway involving PKR-STAT3-PDGF-D-PDGFR-B in injured proximal tubules. The PKR-STAT3-PDGF signaling cascade has been reported in PKR-null mouse embryonic fibroblasts (*Deb et al., 2001*). PDGF-D and PDGFR-B contribute to fibrosis in glomeruli and the tubulointerstitium in experimental animal studies (*Kok et al., 2014*; *Buhl et al., 2020*). Inhibiting this pathway may offer an avenue to reduce kidney injury in HIVAN. Further studies will be needed to confirm whether this mechanism is shared with human HIVAN or other RNA virus-associated kidney diseases.

The present study has limitations. First, the Tg26 mouse model involves a partial HIV-1 transgene that may not recapitulate all aspects of clinical HIVAN. Second, gene expression changes after C16 treatment may include changes secondary to the attenuated renal injury, in addition to the direct effect of C16. Third, we acknowledge the possibility of a non-specific effect of C16 as an inhibitor of PKR (*Chen et al., 2008*; *Lopez-Grancha et al., 2021*; *Cusack et al., 2023*).

In conclusion, by combining single-nucleus RNA-seq and bulk RNA-seq analysis, we identified mitochondrial dysfunction as the central mechanism for proximal tubule injury in the Tg26 HIVAN mouse model. This process was largely reversed by treatment with the PKR inhibitor C16. Further studies of HIVAN-associated mitochondrial dysfunction may lead to targeted therapeutics.

## Materials and methods
### Mice
Hemizygous (Tg26$^{+/-}$) male mice were bred with WT FVB/N female mice to generate Tg26 hemizygous mice. Transgenic mice were identified by PCR genotyping. We studied both male and female mice, aged 6–12 weeks. Mice in treatment groups were matched for sex.

For PKR inhibition treatment, the C16 treatment group mice received 10 µg/kg body weight of C16 (Sigma-Aldrich, St. Louis, MO, USA) dissolved in 0.5% DMSO-PBS (10 ml/kg body weight), administered intraperitoneally three times weekly (*Okamoto et al., 2018*) from 6 weeks to 12 weeks of age. Urine collection (in 24 hr metabolic cages) and body weight measurements were performed at 6 weeks of age before treatment and 12 weeks of age after treatment. Mice were euthanized and plasma, serum, and kidney samples were collected at age of 12 weeks.

### Mouse chemistry measurements
Plasma creatinine was measured by isotope dilution LC-MS/MS at The University of Alabama at Birmingham O'Brien Center Core C (Birmingham, AL, USA). Urine albumin levels were measured using Albuwell M ELISA kits (Ethos Biosciences, Newtown Square, PA, USA). Urine creatinine concentrations were measured using the Creatinine Companion kit (Ethos Biosciences, Newtown Square, PA, USA). Urinary NGAL and KIM-1 were measured by the Mouse Lipocalin-2/NGAL DuoSet ELISA and Mouse TIM-1/KIM-1/HAVCR Quantikine ELISA Kit (R&D Systems, Minneapolis, MN, USA).

### Glomerular and proximal tubule enrichment method
Mice were anesthetized using 2,2,2-trimbromoethanol (Avertin) and the abdominal aorta and vena cava were exposed. After clipping the abdominal aorta distal to renal artery bifurcation, a catheter was inserted into the aorta proximal to an incision above iliac artery bifurcation. After clipping the celiac trunk, superior mesenteric artery, and thoracic aorta proximal to renal arteries bifurcation, a small incision was made in the renal vein to perfuse both kidneys with 1 ml of PBS. Next, kidneys were perfused twice with 10 µl of Dynabeads M-450 Tosylactivated (#14013, Thermo Fisher Scientific, Waltham, MA, USA), to facilitate isolation of glomeruli.

Kidneys were immediately collected, decapsulated, and placed into Hanks' Balanced Salt Solution (HBSS) medium on ice. Kidneys were minced using razor blades and enzymatically digested in 1 ml of HBSS containing 4 mg collagenase A (#10103586001, Sigma, Darmstadt, Germany) and 40 µl DNase I recombinant (E04716728001, Sigma, Darmstadt, Germany) for 30 min at 37°C with shaking at 1500 rpm. Glomerular samples were collected after filtration through a 100 µm strainer, followed by magnetic separation (MPC-S, DYNAL) and three PBS washes. Proximal tubular samples were

collected from the non-glomerular supernatant of the first magnetic separation step. Proximal tubules were isolated by centrifugation through 31% Percoll (17089102, Cytiva, Marlborough, MA, USA)-PBS centrifugation, followed by an additional wash and centrifugation with PBS, following a published protocol (*Iwakura et al., 2019*; *Andargie et al., 2021*).

## Mouse kidney pathological evaluation, ISH, and immunohistochemistry

Mouse kidney tissues were fixed with 10% buffered formalin for 24 hr, embedded in paraffin, and sectioned at 4–5 μm, and stained with hematoxylin and eosin, periodic acid Schiff, and Picrosirius Red. Chromogenic in situ detection of RNA was performed using RNAscope 2.5 HD Reagent Kit (catalog # 322310, Advanced Cell Diagnostics, Biotechne, Minneapolis, MN, USA) with the RNA probes Mm-mt-Co1, Mm-mt-Atp6, dabB (negative control), Mm-Ppib (positive control) (catalog # 517121, 544401, 310043, 313911).

For immunohistochemistry, tissue sections were deparaffinized/rehydrated, antigens were retrieved by citrate buffer, and non-specific binding was blocked. Sections were incubated with primary antibody against phospho-Stat3 (Tyr705) (#9145, 1:100 dilution, Cell Signaling, Danvers, MA, USA), phopho-PKR (Thr 446) (#sc16565, 1:50 dilution, Santa Cruz Biotechnology, Dallas, TX, USA) and PDGF-D (ab181845, Abcam, 1:100 dilution, Cambridge, UK). Sections were processed following ImmPRESS HRP Universal Antibody (horse anti-mouse/rabbit IgG) Polymer Detection Kit and ImmPACT DAB EqV Peroxidase (HRP) Substrate (Vector Laboratories, Burlingame, CA, USA) protocol, and counterstained with hematoxylin.

## Estimation of glomerular podocyte count

PodoCount (*Santo et al., 2022*), a computational tool for whole slide podocyte estimation from digitized histological sections, was used to detect, enumerate, and characterize podocyte nuclear profiles in the glomeruli of immunohistochemically labeled (IHC-labeled) murine kidney sections. Formalin-fixed, paraffin-embedded tissues (2 μm thickness) were IHC-labeled for p57$^{kip2}$, a marker of podocyte terminal differentiation (ab75974, Abcam, Cambridge, UK), and detected with horseradish peroxidase (RU-HRP1000, Diagnostic BioSystems, Pleasanton, CA, USA) and diaminobenzidine chromogen substrate (BSB0018A, Bio SB, Santa Barbara, CA, USA). A periodic acid-Schiff post-stain was applied without hematoxylin counterstain. The tool uses a combination of stain deconvolution, digital image processing, and feature engineering to compute histological podometrics (*Kikuchi et al., 2015*) with correction for section thickness (*Venkatareddy et al., 2014*). In this study, PodoCount was used to assess mean glomerular podocyte count per mouse.

## Immunoblotting

Tissues were lysed in a radioimmunopreciptation assay buffer (Thermo Fisher Scientific, Waltham, MA, USA) containing a protease inhibitor/ phosphatase inhibitor cocktail (#78440, Thermo Fisher Scientific). Lysates were separated by SDS-polyacrylamide gel electrophoresis (gradient gel 4–12%, MOPS buffer) and the proteins subjected to western blotting and blocked for 1 hr in Odyssey blocking buffer (LI-COR, Lincoln, NE, USA). Blots were incubated following the iBind protocol (Thermo Fisher Scientific). Primary antibodies were Phospho-Stat3 (Tyr705) (#9145, 1:2000 dilution, Cell Signaling, Danvers, MA, USA), Stat3 (#9139, 1:1000 dilution, Cell Signaling), β-actin (#47778, 1:5000 dilution, Santa Cruz Biotechnology), Total OXPHOS Rodent (#ab110413, 1:250 dilution, Abcam, Cambridge, UK), VDAC (# 4661, 1:1000 dilution, Cell Signaling). Blots were imaged using the Odyssey infrared scanner (LI-COR, Lincoln, NE, USA).

## Bulk RNA-seq

Mouse kidney outer cortex tissues were dissected and homogenized in QIAzol (QIAGEN, Germantown, MD, USA). Total RNA samples were extracted using RNeasy Plus Universal Kit (QIAGEN) following the manufacturer's protocol including removal of genomic DNA step. RNA samples were pooled and sequenced on NovaSeq6000 S1 flow cell using Illumina TruSeq Stranded mRNA Library Prep and paired-end sequencing with read length 101 bps (2×101 cycles). The samples had 46–72 million pass filter reads and more than 92% of bases calls were above a quality score of Q30. Sample reads were trimmed for adapters and low-quality bases using Cutadapt (*Martin, 2011*). The trimmed reads were mapped to a reference genome (Mouse - mm10). Transcripts were annotated by Ensembl v96 using

STAR aligner. Gene expression quantification analysis was performed for all samples using STAR/RSEM tools. DESeq2 (*Love et al., 2014*) was used for differential expression analysis from raw count data and normalized data were used for GSEA (v4.1.0)(*Subramanian et al., 2005*; *Mootha et al., 2003*). Pathway analysis including upstream regulator analyses were generated using QIAGEN Ingenuity Pathway Analysis (*Krämer et al., 2014*).

### Single-nucleus RNA-seq

Nuclei from frozen mouse kidney outer cortex tissue samples and glomeruli-enriched samples were prepared at 4°C (*Kirita et al., 2020*). Tissue fragments (~8 mm$^3$) were cut by razor blades in EZlysis buffer (#NUC101-1KT, Sigma-Aldrich) and homogenized 30 times using a loose Dounce homogenizer and 5 times by tight pestle. After 5 min of incubation, homogenates were passed through 40 µm filters (PluriSelect, El Cajon, CA, USA) and centrifuged at 500×*g* at 4°C for 5 min. Pellets were washed with EZlysis buffer and again centrifuged at 500×*g* at 4°C for 5 min. Pellets were resuspended in DPBS with 1% FBS and passed through 5 µm filters (PluriSelect) to make final nuclei preparations for loading on to 10x Chromium Chip G (10x Genomics, Pleasanton, CA, USA) and formation of gel beads in emulsion (GEM).

Single-nuclear isolation, RNA capture, cDNA preparation, and library preparation were accomplished following the manufacturer's protocol (Chromium Next GEM Single Cell 3' Reagent Kit, v3.1 chemistry, 10x Genomics). Prepared cDNA libraries were sequenced. Analysis was performed with the Cell Ranger software using the default parameters with pre-mRNA analysis turned on. The reference was built from mm10 reference genome complemented with HIV-1 viral sequences.

### Single-nucleus RNA-seq analysis

SoupX (version 1.5.2) (*Young and Behjati, 2020*) was used to remove ambient RNA, following the default protocol by 'autoEstCont' and 'adjustCounts' functions. Doublets were identified and removed by DoubletFinder (version 2.0.3) (*McGinnis et al., 2019*). Nuclei were filtered out that met any of the following criteria: detected genes <200 or >4000, total RNA count >15,000, or mitochondrial transcripts >20%. Integration of single-nucleus gene expression data was performed using Seurat (version 4.0.5) (*Hao et al., 2021*). After filtering, 57,061 cells remained. Clustering of the combined data used the first 30 principal components at a resolution of 0.6 and identified 25 distinct cell clusters. After removal of 2 doublet clusters, 56,976 cells from 23 clusters were used for downstream analysis. Cell-type identification was done based on the expression levels of known marker genes. Pseudotime analysis of podocytes and proximal tubule cells was performed by using the R package Monocle 3 (version 1.0) (*Trapnell et al., 2014*), considering WT cells as the root state. Cell-cell interaction analysis was performed using CellChat (version 1.5.0) (*Jin et al., 2021*). Pathway analysis including upstream regulator analysis were accomplished through the use of QIAGEN Ingenuity Pathway Analysis (*Krämer et al., 2014*). Cell cycle analysis was performed by converting cell cycle marker genes from *Tirosh et al., 2016*, to mouse orthologs.

### Single-nucleus RNA-seq analysis: comparative analysis with mouse ischemic-reperfusion injury model data reported by *Kirita et al., 2020*

A list of marker genes in injured proximal tubules (NewPT1 and NewPT2) reported by investigators using single-nucleus RNA-seq (*Kirita et al., 2020*) in a mouse ischemic-reperfusion injury model were obtained from the Kidney Interactive Transcriptomics website (https://humphreyslab.com/SingleCell/) for comparison with our data.

### Single-nucleus RNA-seq analysis: PT-Mito cluster in human diabetic kidney disease study reported by *Wilson et al., 2019*

The data from the single-nucleus RNA-seq study of human diabetic kidney disease (diabetic kidney disease n=3; control n=3) reported by *Wilson et al., 2019*, was downloaded from the Gene Expression Omnibus (GEO) using accession number GSE131882. The data was analyzed using Seurat package version 5.0.1 without removing the mitochondrial gene transcripts.

### Mitochondrial copy number measurements

Determination of mitochondrial copy number of mouse kidney tissues was conducted following a published method (.*Quiros et al., 2017*). Genes encoding 16S rRNA and *Nd1*were measured as

mitochondrial DNA (mtDNA) and the *Hk2* gene was measured as nuclear DNA (nDNA). The mtDNA/nDNA ratios in mouse tissues were quantified by SYBR Green assay using QuantStudio 6 (Thermo Fisher Scientific, Waltham, MA, USA).

## Seahorse extracellular flux assay

Seahorse 96-well assay plates (Agilent, Santa Clara, CA, USA) were pre-coated twice with 20 µl/well of 0.01% poly-L-lysine solution (P4707, Sigma, Darmstadt, Germany) and washed twice with PBS, 200 µl/well. Glomerular or proximal tubular samples were plated with EGM-2 medium (CC-3162, Lonza, Walkersville, MD, USA) and placed in a $CO_2$ incubator for 30 min for the attachment.

Seahorse XF RPMI medium, pH 7.4 (103576-100, Agilent, Santa Clara, CA, USA), was used for glomerular samples and Seahorse XF DMEM medium, pH 7.4 (103575-100, Agilent, Santa Clara, CA, USA), was used for proximal tubular samples. Media were supplemented to reach a final concentration of 10 mM glucose, 1 mM sodium pyruvate, and 2 mM L-glutamine. Reagent concentrations used were 2 µM oligomycin, 2 µM FCCP, 0.5 µM rotenone, and 0.5 µM antimycin A (103015-100, Agilent, Santa Clara, CA, USA).

Seahorse Mito Stress Tests were conducted as described (*Okamoto et al., 2018*). Cells were incubated in a $CO_2$-free incubator for 30 min, after replacement of medium. Data were normalized by total nucleic acid content measured by CyQUANT Cell Proliferation (C7026, Thermo Fisher Scientific) and analyzed by Wave 2.6.1 (Agilent).

## Study approval

Mouse experiments were conducted in accordance with the NIH Guide for the Care and Use of Laboratory Animals and were approved in advance by the NIDDK Animal Care and Use Committee (Animal study proposals, K097-KDB-17 and K096-KDB-20).

## Acknowledgements

We thank the Sequencing Facility and Bioinformatics Group (Frederick National Laboratory for Cancer Research [FNLCR], NCI, NIH) for sequencing and informatics support, Drs. Joon-Yong Chung and Stephen M Hewitt (NCI/NIH) for whole slide scanning, Maria Campos (NEI/NIH) for pathological service, Dr. Daria Ilatovskaya (Medical University of South Carolina) for suggestion of Seahorse assay, and Drs. Mark A Knepper (NHLBI, NIH), Gregory G Germino (NIDDK, NIH), and Michael J Ross (Albert Einstein College of Medicine) for scientific suggestions and supports, Dr. Jurgen Heymann for critical manuscript review. This work utilized the computational resources of the NIH HPC Biowulf cluster (http://hpc.nih.gov). Part of this work was presented at American Society of Nephrology Kidney Week 2020, 2021. The content of this publication does not necessarily reflect the views or policies of the Department of Health and Human Services, nor does mention of trade names, commercial products, or organizations imply endorsement by the US Government. This project has been funded in part with federal funds from the National Cancer Institute, National Institutes of Health, under contract 75N91019D00024. The work was also supported by the Intramural Research Program of the NIH, including the National Cancer Institute, Center for Cancer Research, and the NIDDK.

## Additional information

### Funding

| Funder | Grant reference number | Author |
| --- | --- | --- |
| National Institute of Diabetes and Digestive and Kidney Diseases | ZIADK043411 | Jeffrey B Kopp |
| National Cancer Institute | 75N91019D00024 | Cheryl A Winkler |

The funders had no role in study design, data collection and interpretation, or the decision to submit the work for publication.

### Author contributions
Teruhiko Yoshida, Conceptualization, Data curation, Formal analysis, Investigation, Visualization, Methodology, Writing - original draft, Project administration, Writing – review and editing; Khun Zaw Latt, Formal analysis, Visualization, Writing – review and editing; Avi Z Rosenberg, Formal analysis, Writing – review and editing; Briana A Santo, Software, Formal analysis, Writing – review and editing; Komuraiah Myakala, Bryce A Jones, Xiaoxin X Wang, Investigation, Writing – review and editing; Yu Ishimoto, Methodology, Writing – review and editing; Yongmei Zhao, Resources, Formal analysis, Supervision, Writing – review and editing; Shashi Shrivastav, Investigation, Methodology, Writing – review and editing; Xiaoping Yang, Resources, Writing – review and editing; Vincent M Tutino, Pinaki Sarder, Resources, Software, Methodology, Writing – review and editing; Moshe Levi, Resources, Supervision, Writing – review and editing; Koji Okamoto, Conceptualization, Writing – review and editing; Cheryl A Winkler, Resources, Supervision, Funding acquisition, Writing – review and editing; Jeffrey B Kopp, Conceptualization, Resources, Supervision, Funding acquisition, Writing – review and editing

### Author ORCIDs
Teruhiko Yoshida  https://orcid.org/0000-0002-2049-7347
Komuraiah Myakala  https://orcid.org/0000-0003-3233-047X
Xiaoping Yang  https://orcid.org/0000-0003-1263-2710
Moshe Levi  https://orcid.org/0000-0001-6403-2261

### Ethics
Mouse experiments were conducted in accordance with the NIH Guide for the Care and Use of Laboratory Animals and were approved in advance by the NIDDK Animal Care and Use Committee (Animal study proposals, K097-KDB-17 & K096-KDB-20).

Reviewer #1 (Public Review): https://doi.org/10.7554/eLife.91260.4.sa1
Reviewer #2 (Public Review): https://doi.org/10.7554/eLife.91260.4.sa2
Author response https://doi.org/10.7554/eLife.91260.4.sa3

---

## Additional files

### Supplementary files
• MDAR checklist

### Data availability
Original data files and count tables have been deposited in GEO (GSE205060).

The following dataset was generated:

| Author(s) | Year | Dataset title | Dataset URL | Database and Identifier |
|---|---|---|---|---|
| Yoshida T | 2024 | Transcriptomic analysis of HIV-associated nephropathy (Tg26) mouse kidney | https://www.ncbi.nlm.nih.gov/geo/query/acc.cgi?acc=GSE205060 | NCBI Gene Expression Omnibus, GSE205060 |

The following previously published datasets were used:

| Author(s) | Year | Dataset title | Dataset URL | Database and Identifier |
|---|---|---|---|---|
| Wilson PC, Wu H, Kirita Y, Uchimura K | 2019 | The Single Cell Transcriptomic Landscape of Early Human Diabetic Nephropathy | https://www.ncbi.nlm.nih.gov/geo/query/acc.cgi?acc=GSE131882 | NCBI Gene Expression Omnibus, GSE131882 |
| Kirita Y, Wu H, Uchimura K, Wilcon PC, Humphreys BD | 2020 | Cell profiling of mouse acute kidney injury reveals conserved cellular responses to injury | https://www.ncbi.nlm.nih.gov/geo/query/acc.cgi?acc=GSE139107 | NCBI Gene Expression Omnibus, GSE139107 |

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
