## [Editor Report · eLife assessment]

This study presents **valuable** new insights into a HIV-associated nephropathy (HIVAN) kidney phenotype in the Tg26 transgenic mouse model, and delineates the kidney cell types that express HIV genes and are injured in these HIV-transgenic mice. A series of **compelling** experiments demonstrated that PKR inhibition can ameliorate HIVAN with reversal of mitochondrial dysfunction (mainly confined to endothelial cells), a prominent feature shared in other kidney diseases. The data support that inhibition of PKR and mitochondrial dysfunction has potential clinical significance for HIVAN.

---

## [Referee Report · Reviewer #1 (Public Review)]

Summary:

HIV associated nephropathy (HIVAN) is a rapidly progressing form of kidney disease that manifests secondary to untreated HIV infection and is predominantly seen in individuals of African descent. Tg26 mice carrying an HIV transgene lacking gag and pol exhibit high levels of albuminuria and rapid decline in renal function that recapitulates many features of HIVAN in humans. HIVAN is seen predominantly in individuals carrying two copies of missense variants in the APOL1 gene, and the authors have previously shown that APOL1 risk variant mRNA induces activity of the double strand RNA sensor kinase PKR. Because of the tight association between the APOL1 risk genotype and HIVAN, the authors hypothesized that PKR activation may mediate the renal injury in Tg26 mice, and tested this hypothesis by treating mice with a commonly used PKR inhibitory compound called C16. Treatment with C16 substantially attenuated renal damage in the Tg26 model as measured by urinary albumin/creatinine ratio, urinary NGAL/creatinine ratio and improvement in histology. The authors then performed bulk and single-nucleus RNAseq on kidneys from mice from different treatment groups to identify pathways and patterns of cell injury associated with HIV transgene expression as well as to determine the mechanistic basis for the effect of C16 treatment. They show that proximal tubule nuclei from Tg26 mice appear to have more mitochondrial transcripts which was reversed by C16 treatment and suggest that this may provide evidence of mitochondrial dysfunction in this model. They explore this hypothesis by showing there is a decrease in the expression of nuclear encoded genes and proteins involved in oxidative phosphorylation as well as a decrease in respiratory capacity via functional assessment of respiration in tubule and glomerular preparations from these mouse kidneys. All of these changes were reversed by C16 treatment. The authors propose the existence of a novel injured proximal tubule cell-type characterized by the leak of mitochondrial transcripts into the nucleus (PT-Mito). Analysis of HIV transgene expression showed high level expression in podocytes, consistent with the pronounced albuminuria that characterizes this model and HIVAN, but transcripts were also detected in tubular and endothelial cells. Because of the absence of mitochondrial transcripts in the podocytes, the authors speculate that glomerular mitochondrial dysfunction in this model is driven by damage to glomerular endothelial cells.

Strengths:

The strengths of this study include the comprehensive transcriptional analysis of the Tg26 model, including an evaluation of HIV transgene expression, which has not been previously reported. This data highlights that HIV transcripts are expressed in a subset of podocytes, consistent with the highly proteinuric disease seen in mouse and humans. However, transcripts were also seen in other tubular cells, notably intercalated cells, principal cells and injured proximal tubule cells. Though the podocyte expression makes sense, the relevance of the tubular expression to human disease is still an open question.

The data in support of mitochondrial dysfunction are also robust and rely on combined evidence from downregulation of transcripts involved in oxidative phosphorylation, decreases in complex I and II as determined by immunoblot, and assessments of respiratory capacity in tubular and glomerular preparations. These data are largely consistent with other preclinical renal injury model reported in the literature as well as previous, less thorough assessments in the Tg26 model.

Comments on latest version:

The authors have revised the manuscript to acknowledge the potential limitations of the C16 tool compound used and have performed some additional analyses that suggest the PT-Mito population can be identified in samples from KPMP. The authors added some control images for the in situ hybridizations, which are helpful, though they don't get to the core issue of limited resolution to determine whether mitochondrial RNA is present in the nuclei of injured PT cells. Some additional work has been done to show that C16 treatment results in a decrease in phospho-PKR, a readout of PKR inhibition. These changes strengthen the manuscript by providing some evidence for the translatability of the PT-mito cluster to humans and some evidence for on-target activity for C16. It would be helpful if the authors could quantify the numbers of cells in IHC with nuclear transcripts as well as pointing out some specific examples in the images provided, as comparator data for the snRNAseq studies in which 3-6% of cortex cells had evidence of nuclear mitochondrial transcripts.

---

## [Referee Report · Reviewer #2 (Public Review)]

Summary:

Numerous studies by the authors and other groups have demonstrated an important role for HIV gene expression kidney cells in promoting progressive chronic kidney disease, especially HIV associated nephropathy. The authors had previously demonstrated a role for protein kinase R (PKR) in a non-HIV transgenic model of kidney disease (Okamoto, Commun Bio, 2021). In this study, the authors used innovative techniques including bulk and single nuclear RNAseq to demonstrate that mice expressing a replication-incompetent HIV transgene have prominent dysregulation of mitochondrial gene expression and activation of PKR and that treatment of these mice with a small molecule PKR inhibitor ameliorated the kidney disease phenotype in HIV-transgenic mice. They also identified STAT3 as a key upstream regulator of kidney injury in this model, which is consistent with previously published studies. Other important advances include identifying the kidney cell types that express the HIV transgene and have dysregulation of cellular pathways.

Strengths:

Major strengths of the study include the use of a wide variety of state-of-the-art molecular techniques to generate important new data on the pathogenesis of kidney injury in this commonly used model of kidney disease and the identification of PKR as a potential druggable target for the treatment of HIV-induced kidney disease. The authors also identify a potential novel cell type within the kidney characterized by high expression of mitochondrial genes.

Weaknesses:

Though the HIV-transgenic model used in these studies results in a phenotype that is very similar to HIV-associated nephropathy in humans, the model has several limitations that may prevent direct translation to human disease, including the fact that mice lack several genetic factors that are important contributors to HIV and kidney pathogenesis in humans. Additional studies are therefore needed to confirm these findings in human kidney disease.

---

## [Author Response]

The following is the authors’ response to the previous reviews.

**Responses to recommendations for the authors:**

**Reviewer #1 (Recommendations For The Authors):**
The manuscript would be strengthened with the following key revisions mostly having to do with image quality:(1) It is very difficult in Figure 4B to see which nuclei actually have evidence of mitochondrial transcripts. It might be helpful to provide arrows to specific cells and also to provide some estimate of the percentage of cells with nuclear mt-transcripts as measured by ISH compared to the 3-6% of cortex cell estimate seen in the snRNAseq analysis.

As suggested, now we have added arrows to help readers to see the signals in nuclei. The detection threshold of ISH and single-nucleus RNA-seq should be different, and therefore, measuring estimates of PT-Mito by ISH would not be reliable.

(2) The phospho-PKR images provided as evidence of C16 activity (Supplemental Figure 1) are too dim to be very useful. Could brighter images be provided?

We have now adjusted the LUTs of images in Supplemental Figure 1.